# High-Throughput Phenotyping Methods for Breeding Drought-Tolerant Crops

**DOI:** 10.3390/ijms22158266

**Published:** 2021-07-31

**Authors:** Minsu Kim, Chaewon Lee, Subin Hong, Song Lim Kim, Jeong-Ho Baek, Kyung-Hwan Kim

**Affiliations:** 1National Institute of Agricultural Science, RDA, Wanju 54874, Korea; minsu2020@korea.kr (M.K.); wowlek44@korea.kr (C.L.); subinghd@korea.kr (S.H.); greenksl5405@korea.kr (S.L.K.); firstleon@korea.kr (J.-H.B.); 2Department of Crop Science and Biotechnology, Chonbuk National University, Jeonju 54896, Korea

**Keywords:** high-throughput phenotyping, drought, phenomics, breeding

## Abstract

Drought is a main factor limiting crop yields. Modern agricultural technologies such as irrigation systems, ground mulching, and rainwater storage can prevent drought, but these are only temporary solutions. Understanding the physiological, biochemical, and molecular reactions of plants to drought stress is therefore urgent. The recent rapid development of genomics tools has led to an increasing interest in phenomics, i.e., the study of phenotypic plant traits. Among phenomic strategies, high-throughput phenotyping (HTP) is attracting increasing attention as a way to address the bottlenecks of genomic and phenomic studies. HTP provides researchers a non-destructive and non-invasive method yet accurate in analyzing large-scale phenotypic data. This review describes plant responses to drought stress and introduces HTP methods that can detect changes in plant phenotypes in response to drought.

## 1. Introduction

World grain production must more than double by 2050 to provide sufficient bioenergy, feed, and other products in the face of population growth [1]. However, the current increase in crop production is slower than the population growth rate [2]. Furthermore, crops are often devastated by biotic and abiotic stresses, which are aggravated by global warming [3]. Drought is one of the most urgent problems in agriculture due to irregular but frequent changes in precipitation accompanied by increasing temperatures [4]. Numerous attempts have been made to create drought resistant cultivars [5,6]. Following the emergence of next-generation sequencing (NGS) technologies, the physiological responses of crops to drought stress and their underlying molecular mechanisms have been studied using those such as quantitative trait locus (QTL) mapping and genome wide association study (GWAS) [7]. Nonetheless, not many drought-tolerant cultivars have been released due to the breeding bottleneck, i.e., the lack of an accurate, high-throughput phenotyping method, as traditional crop phenotyping methods are labor intensive, time consuming, and subjective. With the advent of phenomics in the form of automatic high-throughput phenotyping (HTP) platforms, it appeared that many high-quality data could now be collected [8]. However, to harness the full benefits of such platforms, it is essential to understand the relationships between the phenotypes quantified by sensors and their physiological relevance. Understanding such relationships is important to optimize these platforms by choosing the methods that match the research goals.

Here, we review current HTP methods, including the specific use of these methods to explore plant responses to drought stress. 

## 2. Plant Responses to Drought Stress

### 2.1. Changes in Water Content

Drought, a major cause of soil moisture deficit, decreases the relative water content (RWC) of plants [9]. Science, The RWC calculation method proposed by Schonfeld et al. [10] was traditionally used to assess the relationship between drought stress and RWC. RWC is used as a measure of plant water status, as it reflects the metabolic activity of tissues and is the most significant indicator of dehydration tolerance. RWC is related to water absorption by the roots and water loss by evaporation [11]. Until recently, studies on changes in RWC in crops under drought stress have been performed using passive analysis methods [12]. Persistent drought causes canopy withering. Drought-sensitive plants wither quickly, but in resistant plants, this process appears to be delayed [13]. 

### 2.2. Changes in Biomass

Plant growth is the result of the production and expansion of daughter cells through cell division. Water deficiency in the plant due to drought stress inhibits water flow from the xylem to extension cells and causes a decrease in turgor pressure. Also, it reduced the rate of cell division and elongation, which ultimately leads to decreased plant growth [14]. The lack of moisture in plants reduces the number, size, and longevity of leaves, thereby reducing the photosynthetic rate. In turn, the low photosynthetic rate leads to a decrease in plant height, stem diameter, and biomass [15]. 

Root diameter, root depth, and root hair development positively correlate with plant vitality under drought stress [16]. The exposure of plants to drought stress reduces the absorption of nutrients in the roots and their transmission from root to shoot [17]. In addition, the density of the underground parts of the plant increases to facilitate water absorption via the development of the main root and root hairs. 

Soil moisture deficit inhibits the shoot growth to prevent water loss while promoting root growth to increase the water uptake [18]. The increase in root biomass enables stable water absorption under soil moisture deficit conditions, allowing the shoot to continue to supply energy for seed production [19]. 

### 2.3. Physiological Changes (Stomatal Closure, Chlorophyll Deficiency, Photosynthetic Rate) in Response to Drought Stress

When roots recognize drought conditions, they transmit signals to the leaves through the xylem, and the plant adapts to the stress via physiological signals. This reaction between the roots and shoots leads to stomatal closure, a basic method to prevent water evaporation. This process suppresses moisture loss in the short term, but in the long term, it reduces biomass production by reducing the rate of photosynthesis and gas exchange between the plant and the atmosphere. An increase in leaf temperature accompanies this process. When this temperature reaches a critical value, it often leads to irreversible leaf tissue damage.

Drought stress interferes with photosynthetic factors including thylakoid electron transport and stomatal control, causing a decrease in photosynthesis [20]. In addition, it interferes with gas exchange in leaves due to decreased leaf expansion, early leaf aging, oxidation of chloroplast lipids, and changes in plastid structure [21]. 

Relative chlorophyll content has a positive relationship with the photosynthesis rate. A decrease in chlorophyll content under drought stress is a typical symptom of oxidative stress and may result from the photo-oxidation and decomposition of photosynthetic pigments. These pigments are essential in plants, as they absorb light and generate energy. However, chlorophyll *a* and *b* are susceptible to water shortage and the reduction in chlorophyll content negatively affects photosynthesis [22].

### 2.4. Yield Changes

Crop yield is the result of the expression and combination of several plant growth factors. Drought stress reduces crop growth and metabolic activity, thereby reducing yield [23]. Poor gas exchange in leaves due to stomatal closure causes a decrease in photosynthetic activity. It severely reduces the harvesting characteristics of crops due to the insufficient production of photosynthetic products [15]. In legumes, drought stress at reproductive growth stages like flowering significantly reduces the pod numbers and yields [24]. 

## 3. Htp Methods

Since the first report on the use of spectral and thermal radiance in an attempt for high throughput phenotyping for drought tolerance in 2011 by Winterhalter et al. [25], HTP tools became a familiar tool for assessing drought tolerance. Abiotic stress resistance is influenced by a variety of genes with minor effects. Therefore, a method is required to accurately confirm even small changes in each phenotypic measure that contribute to resistance. In addition, such a method must be conducted non-destructively for the quantitative evaluation of stressed plants. One method that satisfies all of these requirements is HTP method [26]. The following section, we will discuss the different types of HTP tools and their applications for assessing drought tolerance in plants (Table 1).

In this review, we used the search engine Scopus (https://www.scopus.com/search/form.uri?display=basic#basic, accessed on 28 June 2021) to investigate the flow of mass testing of drought traits between 2007 and April 2021. The search was conducted using keywords including high-throughput, phenotyping, and drought, and we identified 248 articles. The search was subdivided wheat, rice, barley, maize, sorghum, and soybean (Figure 1D). Between 2007 and 2010, fewer than five articles were published, and more than six articles were published each year from 2011 to 2021 (Figure 1A). From 2018 to 2020, more than 30 papers were published every year, representing an increase in the number of studies on drought HTP testing (Figure 1A). The types of journals were agricultural and biological sciences (49%), biochemistry, genetics and molecular biology (21%), engineering and computer science (5%), and environmental science (4%) (Figure 1B). Among the document types, 68% of the published articles were research articles, 14% were review articles, and 10% were book chapters. Drought is an important problem worldwide. Researchers from at least 68 countries studied this topic during this period, including researchers in approximately 190 national research institutes, companies, and universities, including 71 in the United States, 39 in India, 34 in Australia, 34 in France, and 21 in Germany. This literature survey also indicated that investigations dealing with drought stress were active in major developed and agriculture-oriented countries (Figure 1C).

### 3.1. RGB Imaging

The method using RGB camera is the most widely used system to measure the morphological properties of plants due to its cost-effectiveness and ease of installation [56]. Unlike consumer cameras, RGB cameras contain an infrared blocking filter (VIS camera) that detects light wavelengths between 400 and 700 nm. The VIS camera uses red, green, and blue color sensors to measure the color of each pixel. The pixel values of plants identified by the image processing algorithm are utilized to collect morphological or color information [57]. 

Laboratory experiments primarily involve the use of fixed facilities to minimize experimental variables under controlled environmental conditions. Such studies also enable detailed analysis of the responses of individual plants to stress conditions and the acquisition of image information throughout the experiment. Ge et al. [27] extracted plant pixels from RGB images in two levels of water application and used them to establish the correlation with destructively measured shoot fresh weight, dry weight, and leaf area. Plant projected area extracted from side view of maize RGB images can be accurately related to destructively measured plant shoot fresh weight, shoot dry weight, and leaf area at the early growth stage. Neilson et al. [28] estimated the overall biomass of the plant and compared it with actual plant size determined on the destructively harvested plants. Strong positive correlations were detected between projected leaf area and shoot biomass in the water-limiting experiment. Also, the greenness of the leaves was estimated by converting the images from the RGB to the Hue Saturation Intensity (HIS) color management system. In field experiments, the sensor method used depends on the method employed to collect the visualized parameters of the plant response. Field studies are performed when row analysis or large-area experiments are required rather than individual plant analysis. In field experiments, phenotypic analysis is performed by attaching sensors to vehicles or aircraft to cover a large area. The RGB sensor used in an unmanned aerial vehicle (UAV) can acquire high-resolution color space information quickly. Based on that information, can collect various vegetation indices. Bhandari et al. [31] and Francesconi et al. [32] computed canopy features such as canopy cover and canopy height using UAV in wheat. These study shows high-throughput UAV data can be used to monitor the drought effects on wheat growth and productivity. Otherwise, RGB camera can be attached to fixed facilities to reduce errors in data due to vehicle movement. Becker and Schmidhaler [30] used an RGB sensor at the top of a field drought treatment facility to distinguish between wheat and ground cover and estimated the correlation between yield and plant growth rate under drought stress using RGB image extraction.

Changes in plant biomass under drought stress can be identified and analyzed pixel-by-pixel using RGB images. Biomass content inference using pixel counts has shown a high correlation in various crops [29]. Therefore, RGB images are helpful in predicting changes in the growth rates of plants under drought stress, making it possible to analyze whole plants or specific parts of a plant. In addition, color analysis can confirm leaf wilting and chlorophyll deficiency due to drought stress [32]. The color information is obtained in this manner, and additional information is used to determine the vegetation index. In field experiments, since most RGB sensors are attached to UAVs, the acquired images are often limited to the upper image [31,32]. Therefore, such images are used for vegetation index analysis, such as leaf area index, canopy and chlorophyll content rather than accurate biomass analysis. They are used as a primary indicator to distinguish between plant area and soil area based on color information. The phenotypic information obtained through a series of processes is also used to search for candidate genes underlying a particular phenotype. Campbell et al. [29] estimated daily shoot biomass and soil water content using HTP platform and modeled shoot growth. After that, several candidate genes were identified by combining a genome-enabled growth model. RGB imaging is the most basic HTP analysis method. However, it is difficult to use this system alone in combination with UAV because the visible image provides only morphological information, and it is challenging to separate leaves and soil of similar colors during the image segmentation process. Therefore, RGB sensors are mainly used as base sensors in combination with additional sensors of other types.

### 3.2. Near-Infrared Imaging

In many studies, the green areas of plants showed the highest reflectance at near-infrared wavelengths between 700 and 1400 nm. By contrast, soil reflects very little near-infrared light, and unhealthy plants reflect more red light than healthy plants. In addition, leaves that have recently absorbed water cause scattering of near-infrared wavelengths. This property of near-infrared imaging (NIR) is used to confirm the transformation of plants under drought stress [58]. Among the various absorption bands, the absorption rate is highest in the spectral range of 1400 to 1450 nm and is highly correlated with the moisture content of plants. Chen et al. [36] investigated the dynamics of water content of the drought responses of 18 different barley cultivars using NIR imaging. Plants showed a rapid decrease of the NIR signal after about drought stress and recover signal after re-watering. Das et al. [37] identified water absorption bands using partial least squares regression (PLSR) model from NIR image. NIR can provide an effective means for real-times and non-invasive monitoring of leaf water content in crop plants. The plant water content estimated by NIR is also used to identify candidate gene regions. El-Hendawy et al. [38] used a combination of NIR and short-wave infrared (SWIR) imaging as a method for selecting spring wheat genotypes with superior yields under drought stress and could identify the drought-tolerant parent and several RILs. In other words, the measurement of reflectance in a specific wavelength band using NIR enables rapid analysis of plant water content, allowing this method to detect drought stress in plants quickly.

### 3.3. Hyperspectral Imaging

Hyperspectral sensors detect hundreds of thousands of bands per pixel, spanning the visible (400–700 nm), NIR (700–1000 nm), and SWIR (1000–2500 nm) regions [59]. Although the physical complexity of the collected data requires high-performance analytical computers, sensitive detectors, and large data storage capacity, images can be acquired at high resolution with narrow spatial coverage to differentiate responses to different stresses. Hyperspectral images can detect early drought stress symptoms, which are not yet visible to the naked eye. The use of the appropriate spectral analysis technique makes it easy to evaluate the responses of plants to drought stress.

Stationary facilities used in the laboratory can produce normalized difference vegetation index (NDVI) values using various hyperspectral wavelength ranges that can be separated and subdivided based on plant structure. This technique is effective for checking the water content and canopy reflectance of each part of the plant. Ge et al. [27] calculated NDVI of maize using 670nm and 770nm wavelength and classified the plant into stem and leaves using 1160nm wavelength. The information obtained through hyperspectroscopy is limited to information about the interactions between light, plants, and soil materials. However, transpiration is not directly related to spectral reflection, but this process can be detected by adding a sensor that uses a different physical principle. Therefore, the hyperspectral sensor in the field is primarily used in conjunction with other sensors.

Most hyperspectral sensors used in the field are attached to UAVs, and some are intermittently attached to vehicles. The combination of a hyperspectral camera and UAV increases throughput and enhances measurement objectivity. The validity of the vegetation index obtained via UAV hyperspectral imaging was confirmed by a correlation between the spectrum acquired from the ground and UAV [58]. Rischbeck et al. [39] calculated thirteen spectral indices derived from hyperspectral and thermal data and tested their correlation with grain yield under drought stress. Tattaris et al. [41] identified the correlation between plant wilting and yield under drought stress using NDVI and green normalized difference vegetation index (GNDVI). As such, hyperspectral has the advantage of extracting desired wavelengths from a wide wavelength. Currently, the hyperspectral of the HTP platform is focused on combining multi spectral to identify different vegetation indexes. Depending on which wavelengths are used, the vegetation index of a particular plant organ can be effectively identified [45,60]. Hyperspectral data were also used to introduce genomic estimated breeding combined with molecular marker. Trachsel et al. [43] improved pre-harvest predictions of grain yield combined with molecular marker and hyperspectral data above 700 nm wavelength. A common approach to the hyperspectral-based estimation of plant properties under drought stress is to utilize the vegetation index, which is defined as a linear combination or ratio of reflectance at several single wavelengths. The physiological and biochemical responses of vegetation to drought stress are compared, including changes in the photosynthetic apparatus, water content, plant organs, and yield. In short, hyperspectral imaging can analyze changes in plants under drought stress with various vegetable indices.

### 3.4. Fluorescence Imaging

The photosynthetic rate is an indicator that can confirm the effects of drought stress on the phenotypes of crops. Fluorescence sensors can be used to examine the photosynthetic efficiency of the crop being tested by measuring chlorophyll fluorescence, i.e., the emission of unnecessary energy by the plant in the form of fluorescence [61]. A visible or ultraviolet (UV) light is used to stimulate the plant, and fluorescence is imaged with a charge-coupled device (CCD) cameras [62]. UV in the 340–360 nm range produces two types of fluorescence: red + far-red fluorescence and blue-green fluorescence. Fluorescence emitted in four spectral bands is collected using a multicolor fluorescence imaging principle, each represented by a blue (F440), green (F520), red (F690), and near-infrared (F740) wavelength [63].

Fluorescence and chlorophyll content are key indicators of the metabolic states of plants. Measuring these parameters is an effective way to identify changes in photosynthesis and pigment ratios caused by drought. Bürling et al. [46] analyzed drought-induced changes of blue, green and far-red fluorescence of four different wheat cultivars. Under drought stress, blue-to-far-red fluorescence ratio (BFRR) was significantly increased in all wheat cultivars, but the cultivars previously classified as more tolerant of drought had a stronger BFRR modification than sensitive cultivars. Kim et al. [64] analyzed F_0_, Fm, and Fv values to determine photosynthetic efficiency according to chlorophyll fluorescence in rice and drought tolerant mutant showed higher Fv, Fm, Fm/F_0_ and Fv/Fm than wild types. Chlorophyll fluorescence has also been quantified in a specific plant part, including the spike, leaf, and stem [49] or in the top-view and side-view of a plant [36,50]. The optimal photosystem II (PS II) efficiency and parameters for the HTP experimental procedure must be identified [51]. McAusland et al. [52] proposed an experimental method for observing photosynthesis and the photoprotection of crops in various light environments through fluorescence analysis. Likewise, in the field, photosynthetic efficiency can be analyzed by measuring the chlorophyll fluorescence of a target plant using a portable fluorescence meter in a drought-stress or well-watered environment [53,54,55]. 

Chlorophyll fluorescence helps monitor linear electron transport, which is closely related to CO_2_ uptake during photosynthesis [65]. Drought stress affects the chlorophyll content and its composition, thus resulting in reduced photosynthesis. Therefore, the fluorescence imaging method is a useful analytical method for identifying physiological changes, such as leaf gas exchange, photosynthesis rate, stomatal conductance, and CO_2_ uptake rate, in plants under drought stress. However, most experiments evaluating photosynthetic efficiency via fluorescence analysis have been conducted in the laboratory because fluorescence analysis requires prior steps such as dark adaptation and is sensitive to changes in external light, making it technically challenging to perform this analysis in the field [66].

### 3.5. Thermal Imaging

The thermal imaging method (also known as infrared thermal imaging method) uses radiation emitted by an object to generate an image, which increases as the temperature of the object increases above absolute zero. Thermal sensors can use visualized image data to detect changes in plant temperature caused by transpiration due to stomatal closure. Therefore, thermal imaging can measure temperature-related features such as water content, transpiration rate, and stomatal conductance through model-based estimation [67].

An experiment examining drought resistance using a thermal sensor was performed in the field to analyze the temperature differences between the plants and the plot under drought stress. Sankaran et al. [40] collected thermal imaging using octocopter UAV with thermal camera in the terminal drought experiment. Prior to feature extraction, background removal was performed using thresholding to eliminate soil interference in the extracted canopy features and calculated the average temperature of each plot. When the temperature data within an image were compared and there was a significant correlation between the seed yield and the canopy temperature. The correlation of drought stress with biomass and yield was confirmed, despite the dynamic data, sensitivity to environmental factors such as wind and clouds. Relevant genomic regions were uncovered, and extreme genotypes for canopy temperature were identified. Kaler et al. [68] identified genomic regions associated with canopy temperature and to identify extreme genotypes. Aerial thermal infrared image analysis was implemented to evaluate canopy temperature. Association mapping was conducted using canopy temperature data and identified 52 SNPs significantly associated with canopy temperature. The canopy temperature obtained from the thermal image was used to estimate the crop water stress index (CWSI). CWSI is used as a standard for selecting drought-resistant varieties or as an index to classify the responses of plants based on the degree of water stress [39,42]. In all studies, plant temperature was correlated with NDVI, and drought stress increased the plant temperature. In addition, drought-resistant genotypes had relatively low temperatures. The addition of thermal cameras can improve the phenotyping of crops for drought adaptation.

Transpiration is a process by which the water absorbed by a plant is converted from liquid to vapor and delivered to the atmosphere, resulting in a decrease in plant canopy temperature [69]. Water stress causes partial or total stomatal closure in the plant, thus reducing the transpiration rate and increasing the canopy temperature [68]. Thermal imaging provides information related to stomatal conductance by allowing the researcher to check the increase in temperature in the plant quickly. This method makes it possible to evaluate changes in the transpiration rates of plants and the degree of drought stress.

## 4. Future Perspective

Crop breeding has evolved from the traditional breeding by subjective datum method to phenotypic-genotypic integrated breeding by the advancement of the sequencing method of crops. Now, the breeding faces with new era with the advent of phenomics. It enables breeders to phenotype numerous of samples in an accurate manner. If it combines with NGS technology, breeders could be able to associate a lot more phenotypes with accordant genotypes. 

In recent years, there are even more advanced attempts in the phenomics area; computing methods such as machine learning (ML), deep learning (DL), and artificial intelligence (AI) are combined with the high throughput phenotyping analysis to predict the performances of the breeding population of various crops (Table 2). ML, DL, and AI are inherently multidisciplinary approaches to data analysis, which are typically robust when large amounts of data are dealt with [42]. This trend will be accelerated in the breeding sectors.

## 5. Concluding Remarks

Water is an essential element required for the early and continuous growth of crops. Drought inhibits the steady growth of plants, ultimately causing a decrease in yields. Hence, developing drought-tolerant varieties through plant breeding is essential and considered to be a promising approach. Studies on drought stress have been conducted using various omics approaches; one approach that has recently attracted attention is phenomics.

The demand for precise and rapid phenotypic analysis has led to the development of HTP methods. Because phenotypes are influenced by various factors, representing the results of different reactions in plants, multiple sensors such as RGB, fluorescence, infrared, and hyperspectral sensors are required for phenotypic analysis. RGB images are suitable for detecting morphological changes in plants. Although the ability of RGB sensors to detect visible phenotypes such as reduced biomass and wilting of leaves is excellent, there is a limit to our understanding of the underlying physiological processes. Fluorescence, infrared, and hyperspectral parameters that can be determined using other sensors can provide information about physiological features such as photosynthetic rate and water content. It is important to understand the range of capabilities of each sensor, and ultimately, all sensor information must be integrated to produce indicators describing the resistance phenotype of a plant. Finally, the appropriate remote field testing or indoor expression system should be selected based on particular the environmental conditions. Collectively, this article reveals the importance of HTP methods in assessing the drought phenotypes, which expected to fasten the breeding programs for drought-tolerant crops.

## Figures and Tables

**Figure 1 ijms-22-08266-f001:**
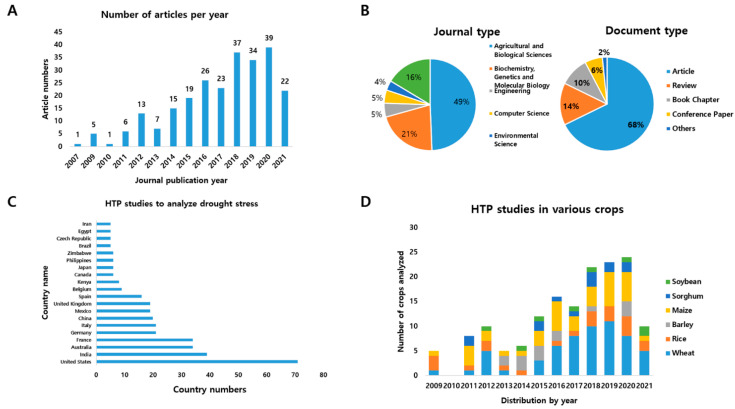
Status of high-throughput phenotyping and drought stress studies performed in 2007–2021. (**A**) The number of journals per year. (**B**) Journal and document type. (**C**) HTP studies to analyze drought stress. (**D**) HTP studies in various crops.

**Table 1 ijms-22-08266-t001:** Sensors used for high-throughput phenotyping of plants under drought stress.

Sensor	Platform	Trait	Advantages	Limitations	References
RGB	Laboratory, fixed	Shoot biomass, leaf/shoot area, identify aging through color analysis, growth rate	Environmental control possible, relatively affordable	Only provides morphological information	[27,28,29]
	Field, fixed	Distinguish ground cover and plants	Quickly obtain information on breeding rows for large populations	Image blur and area mismatch errors occur due to changes in light conditions	[30]
	Field, aerial	Leaf area index (LAI), canopy length, wilting characteristics, yield estimation			[31,32]
	Field, vehicle	NDVI, yield estimation			[33]
	Field, portable	Root biomass, canopy color, canopy length			[34,35]
NIR	Laboratory, fixed	Water content, canopy spectrum reflectance	Low water reflectance, simple use	Limited use with narrow reflectance range	[36,37]
	Field, portable	Canopy reflectance			[38]
Thermal	Field, aerial, vehicle	Canopy temperature	Able to quickly analyze increases in plant temperature	Difficulty in distinguishing plants	[39,40,41,42]
Hyperspectral	Laboratory, fixed	Leaf water content, canopy spectrum	Precise data detection possible using multiple bands	Difficulty in selecting and calibrating specific bands	[27,39]
	Field, aerial	Plant height, flowering date, yield estimation, NDVI,			[40,41,43,44,45]
	Field, vehicle	Canopy reflectance, solar radiation			[39,45]
Fluorescence	Laboratory, fixed	Chlorophyll fluorescence, gas exchange, transpiration, stomatal conductance	Automatic and rapid measurement of photosynthetic rate	Difficult to use in the field, requires dark adaptation	[36,46,47,48,49,50,51,52]
	Field, portable	Chlorophyll fluorescence			[53,54,55]

**Table 2 ijms-22-08266-t002:** Big data analysis technology using artificial intelligence (AI).

Model	Method	Sensor for Used Data	Crop	Trait	Performance	Reference
Machine learning	K-means clustering	RGB	*Macrotyloma uniflorum*	plant shoot length, height, flowering percentage, pod, pod length, number of seeds	clustering complete	[70]
		RGB, NIR	*Brassica napus*, *Camelina sativa*, *Fabaceae*, *Cicer arietinum*	flowering detection UAV	>72%	[71]
	RF ^1^	RGB	*Zea mays*	identify location and growth rate of silk	>86%	[72]
	RF, k-nearest neighbor, univariate permutation test	scanner	*Vicia faba*	root architecture	RF produce the best performance	[73]
	SVM ^2^	RGB, IR, HS	*Beta vulgaris*	water, nitrogen, and weed stress	>93, 76, and 83%	[74]
		RGB, IR	*Glycine max*	canopy wilting	>76%	[75]
	SVM, RF, ANN ^3^	3D laser scanning	*Cicer arietinum*	evapotranspiration (ET)	SVM produced the best results	[76]
Deep learning	CNN ^4^	RGB, multispectral	*Glycine max*	yield estimation	>78%	[77]
		RGB	*Zea mays*	water stress	>88%	[78]
		RGB	*Triticum*	root architecture	r^2^ = 0.64–0.99	[79]
		HS	*Zea mays*	relative water content (RWC)	>87%	[80]

^1^ Random forest. ^2^ Support vector machines. ^3^ Artificial neural networks. ^4^ Convolutional neural network.

## Data Availability

Not applicable.

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
