# Peer review of "High-Throughput Phenotyping Methods for Breeding Drought-Tolerant Crops"

_ijms, 2021, doi:10.3390/ijms22158266_

Round 1

Reviewer 1 Report

The manuscript describes the use of high-throughput phenotyping in plant breeding for drought tolerance. This direction is especially actual in the light of global climate change. The authors report the current HTP systems and discuss future perspectives, but there are some comments.

At the beginning of Section 3, it is desirable to report on the first studies on the application of HTP for assessing plants for drought tolerance

Section 3 should be supplemented with a subsection about laser scanning for creation of 3D models of plant architecture

In Section 4, it is desirable to briefly describe the artificial intelligence methods listed in Table 2

It is desirable to mention the specific techniques used in the evaluating of drought tolerance: platforms for assessing transpiration dynamics and water use efficiency using weight loss of pots, transparent containers for studying the root system, etc.

Ref. 35, 91, 111, 127, 131 are incomplete.

Ref. 43, 45, 50, 54, 59 are very old.

Reviewer 2 Report

Kim and colleagues' manuscript is very interesting and a hot topic in the agricultural and botanical fields. The authors propose a review of the literature on http systems for monitoring drought resistance. After a description of the biochemical and physiological aspects of plants to combat drought, the authors discuss how to acquire information on the health of the plant. The first concerns the acquisition of morphological information, the second concerns information on the plant's response to light, and finally the transpiration processes.

Although I found the manuscript interesting, in my opinion there would be no references to the strictly molecular aspects. Indeed, the technologies that the authors discuss measure macroscopic parameters. IJMS focuses interest on strictly molecular aspects that are not discussed in this review. Are there systems that evaluate molecular parameters directly associated with drought? Gene or protein expression?

For this reason, this review would be more suitable for a botanical or agricultural journal.

Scientific names of plant species should be included in Table 2.

The literature should be integrated with some recent works on the subject:

Naservafaei, S. et al. Biological Response of Lallemantia iberica to Brassinolide Treatment under Different Watering Conditions. Plants 202110, 496. doi: 10.3390/plants10030496

Reza Yousefi, A. et al.  Germination and Seedling Growth Responses of Zygophyllum fabago, Salsola kali L. and Atriplex canescens to PEG-Induced Drought Stress. Environments 20207, 107. doi: 10.3390/environments7120107

Khaleghnezhad V et al. Concentrations-dependent effect of exogenous abscisic acid on photosynthesis, growth and phenolic content of Dracocephalum moldavica L. under drought stress. Planta. 2021 May 25;253(6):127. doi: 10.1007/s00425-021-

Reviewer 3 Report

The MS shows a number of weaknesses so that in the current version it should be rejected.

In fact, despite the title there is a lack of examples able to effectively explain the relationship between HTP and breeding, while it is only mentioned in the "Future perspective" the importance of implementing HTP with NGS technology or computing technologies. In addition, the MS appears to be a mere list of alternative imaging techniques with little attention to specific peculiarities or higher/lower efficiency in relation to HTP.

Moreover:

Chapter 2 should be shortened and references reduced, keeping only the most recent ones (more than 50 references to confirm the effects of drought are too many; It is also unnecessary to report the formula for calculating the RWC).

It is necessary to indicate up to which month of 2021 the Scopus search was carried out.

It is important in the text not to confuse technologies with techniques.

If the Scopus search was done for wheat, rice, barley, maize, sorghum, and soybean, also table 2 should in principle cover the same species.

Table 2 is included in the "Future perspective" section, but the various data / indications must be included in the different imaging paragraphs.

The use of Italics for species names is mandatory.

The number of citations seems excessive.

Round 2

Reviewer 2 Report

The manuscript has improved considerably; the authors have finalized their paper following the indications suggested by the reviewers

I am very satisfied with the corrections and additions made by the authors.

In my opinion, the manuscript can now be published.

Reviewer 3 Report

I'm sorry but the authors didn't edit the MS enough to change my mind.
They ave very little shortened the introductory part and continue to lack clear examples regarding the use of HTP for breeding.
Finally, table 2 shifts the focus from HTP / imaging to computational models moreover inserting references already included in table 1.
Therefore, for a lack of linearity and clarity in the exposition, I think that the review deserves to be not accepted.

Round 3

Reviewer 3 Report

I am sorry but my opinion does not change because there are no improvements in the current version of the MS, nor have the Authors responded adequately to previous comments.
The review is essentially a list of imaging methods, there are no examples to illustrate their relevance to breeding and the range of capabilities of each sensor is not discussed.
